# Liposomal Amphotericin B for Treatment of Leishmaniasis: From the Identification of Critical Physicochemical Attributes to the Design of Effective Topical and Oral Formulations

**DOI:** 10.3390/pharmaceutics15010099

**Published:** 2022-12-28

**Authors:** Frédéric Frézard, Marta M. G. Aguiar, Lucas A. M. Ferreira, Guilherme S. Ramos, Thais T. Santos, Gabriel S. M. Borges, Virgínia M. R. Vallejos, Helane L. O. De Morais

**Affiliations:** 1Department of Physiology and Biophysics, Institute of Biological Sciences, Universidade Federal de Minas Gerais, Belo Horizonte 31270-901, MG, Brazil; 2Faculty of Pharmacy, Universidade Federal de Minas Gerais, Belo Horizonte 31270-901, MG, Brazil

**Keywords:** liposomes, amphotericin B, leishmaniasis, oral route, PEGylation, topical route

## Abstract

The liposomal amphotericin B (AmB) formulation, AmBisome^®^, still represents the best therapeutic option for cutaneous and visceral leishmaniasis. However, its clinical efficacy depends on the patient’s immunological status, the clinical manifestation and the endemic region. Moreover, the need for parenteral administration, its side effects and high cost significantly limit its use in developing countries. This review reports the progress achieved thus far toward the understanding of the mechanism responsible for the reduced toxicity of liposomal AmB formulations and the factors that influence their efficacy against leishmaniasis. It also presents the recent advances in the development of more effective liposomal AmB formulations, including topical and oral liposome formulations. The critical role of the AmB aggregation state and release rate in the reduction of drug toxicity and in the drug efficacy by non-invasive routes is emphasized. This paper is expected to guide future research and development of innovative liposomal formulations of AmB.

## 1. Introduction

Leishmaniasis, a group of diseases caused by *Leishmania* spp. and transmitted by the bites of female phlebotomine sandflies, is among the top ten neglected tropical diseases worldwide. Distinct species of *Leishmania* cause different clinical manifestations, including cutaneous leishmaniasis (CL), mucocutaneous (MCL) and visceral leishmaniasis (VL). VL is principally caused by *L. donovani* in Asia and Africa and by *L. infantum* in the Mediterranean Basin, the Middle East, central Asia and South and Central America. In contrast, CL is caused by *L. major*, *L. tropica* and *L. aethiopica* in the Old World and *L. amazonensis*, *L. mexicana*, *L. braziliensis* and *L. guyanensis* in the New World. Risk factors for progression of VL and increased spread in all transmission settings include malnutrition, genetic factors, population movement, other infectious diseases and immune suppression, notably HIV infection. The epidemiology of CL in the Americas is very complex, with variations in transmission cycles, reservoir hosts, sandfly vectors, clinical manifestations and response to therapy, and multiple circulating *Leishmania* species in the same geographical area [1,2,3,4,5]. In 2020, more than 90% of the VL cases reported to WHO were in Brazil, Ethiopia, Eritrea, India, Iraq, Kenya, Nepal, Somalia, South Sudan and Sudan, and of the 10 countries with the highest number of cases of CL, 4 are in the Americas: Brazil, Colombia, Nicaragua and Peru. There are more than 12 million infected people, 0.9 to 1.6 million new cases and between 20,000 and 30,000 deaths each year, and 350 million people at risk of infection [1,4].

The treatment of leishmaniasis depends on several factors, including type of disease, concomitant pathologies, parasite species and geographic location [6]. Leishmaniasis is a treatable and curable disease, which requires an immunocompetent system due to parasite multiplication inside the macrophages and the dependence of parasite clearance on the activation status of the host cell. Thus, the risk of relapse exists if immunosuppression occurs. Antimonials, amphotericin B (AmB), pentamidine isethionate and miltefosine constitute the therapeutic arsenal available for systemic treatment of leishmaniasis [7]. Pentavalent antimonials are the oldest drugs available and are still considered first-line treatments against most forms of leishmaniasis in several developing countries [7]. However, their adverse effects—cardiotoxicity, particularly evident in HIV–VL co-infection; renal failure; and pancreatitis—represent limitations of this treatment modality [5,8]. In addition to drug toxicity, a major challenge in the treatment of leishmaniasis is that traditional antileishmanial drugs face specific difficulties penetrating inside the macrophages to reach parasites. This context has stimulated the search for strategies to target and improve drug delivery to the host cell [9], as well as to enhance drug efficacy through host-directed therapy [3,10].

Due to the limited therapeutic options for VL in HIV co-infected patients, until now, the recommended treatment has been monotherapy with AmB as deoxycholate or its lipid formulations. However, the WHO expert committee recommends that countries adopt innovative policies using combination regimens, which have the potential advantages of better efficacy, lower overall dose and better compliance and treatment completion rates, which reduce adverse effects, the probability of selection of drug-resistant parasites and the costs, as well as increase health service efficiency [4,7].

Despite the efficacy of AmB deoxycholate against VL, its use is accompanied by dose-limited toxicities, such as fever, hypokalemia, myocarditis and mainly nephrotoxicity, which demand hospital monitoring [3,11]. To improve the therapeutic index of AmB, three lipid-associated formulations were developed: AmB lipid complex (ABLC), liposomal AmB (LAmB) and AmB colloidal dispersion (ABCD). The lipid composition of all three of these preparations differs considerably and contributes to substantially different pharmacokinetic parameters [7,12,13,14]. In general, lipid formulations have similar efficacy to that of AmB deoxycholate, but with less toxicity [7]. Among the lipid formulations, liposomal AmB, called AmBisome^®^, has a lower incidence of adverse reactions, notably nephrotoxicity and infusion-related reactions. In the mouse model of VL, AmBisome^®^ and ABLC showed higher anti-leishmanial activities when compared to AmB deoxycholate. In the CL mouse model, AmBisome^®^ was found more effective [15]. Now, AmBisome^®^ is recommended as a first-line drug in VL patients in endemic areas, such as India, as well as in the Americas [7,12,16].

While several generic versions of AmBisome^®^ have been approved after expiration of its patents, many were subsequently withdrawn from the market due to product safety concerns [17]. This problem was attributed to the use of different manufacturing processes and their influence on the final formulation characteristics. It also illustrates the challenge posed by the large-scale manufacturing of a highly complex AmB-liposome nanosystem and points to the need to better understand and control the physical state of the drug in the liposome membrane.

Unfortunately, the widespread use of AmBisome^®^ and its generics remains limited by their low stability at temperatures higher than 25 °C and the need for parenteral administration [14,18,19]. Furthermore, AmBisome^®^ showed moderate efficacies in New World CL, with a cure rate lower than 80% [20,21,22]. Thus, efforts have been devoted to improving the efficacy of liposomal AmB against CL and achieving topically and orally active liposomal formulations.

This article first provides an update of our current knowledge on the physicochemical and pharmacological features of AmB, its physical state in liposome membranes and the factors that govern the toxicity and therapeutic efficacy of liposomal formulations against VL and CL. It then reviews the recent advances in the development of more effective liposomal AmB formulations, including topical and oral liposome formulations.

## 2. Physicochemical Characteristics, Mechanisms of Action and Toxicity of AmB

### 2.1. AmB Physicochemical Characteristics

AmB has a complex chemical structure composed of a hydrophilic polyhydroxyl chain, a lipophilic polyene hydrocarbon chain and a mycosamine moiety containing a free amino group (Figure 1) [23,24].

AmB is an amphoteric molecule that presents two ionizable groups: a carboxyl (that can be negatively ionized, pKa = 3.72) and an amine (that can be positively ionized, pKa = 8.12) [25]. When just one group is ionized (pH values below 2 or above 10), AmB is soluble in aqueous media (0.1 mg/mL), but when both groups are ionized (pH values between 6 and 7), AmB exhibits a very low solubility. Regarding organic solvents, AmB is known to be soluble in polar solvents, such as dimethylsulfoxide (DMSO) (30–40 mg/mL) and dimethylformamide (DMF) (2–4 mg/mL) [26].

In water at neutral pH values (around 7.4), AmB dimers are formed by apposition of the hydrophobic moieties of the two monomers, above the AmB critical aggregate concentration (0.001 mg/mL) [27]. AmB dimers continue to self-associate as AmB concentration is increased in a specific medium and/or temperature is increased, forming structures that are known as aggregates.

The AmB monomer was observed after dilution of AmB in water at extreme pH values (below 2 or above 10); dilution in organic solvents, such as DMSO and DMF; or immobilization in drug delivery systems, such as cyclodextrins [28] and nanoemulsions [29]. Regarding AmB oligomers (dimers), those were reported in deoxycholate micelles (commercial product, Fungizone^®^). Regarding higher-order aggregates, they can be obtained when deoxycholate AmB is heated at 70 °C for 20 min or at concentrations above 0.001 mg/mL in an aqueous medium [30,31].

It is possible to distinguish monomeric, oligomeric and aggregated forms of AmB through different physicochemical techniques, such as ultraviolet/visible (UV/Vis) electronic absorption, circular dichroism (CD), fluorescence spectroscopies, powder X-ray diffraction (PXRD), differential scanning calorimetry (DSC) and average size measurements.

#### 2.1.1. UV/Vis Electronic Absorption

The electronic absorption spectrum of AmB is highly sensitive to the state of aggregation of the molecule. Monomeric AmB presents absorption peaks at 363–365, 383–384 and 406–409 nm. When AmB begins to aggregate through hydrophobic interactions, forming oligomers, a very strong absorption peak is detected at 328–340 nm. When oligomers are heated, forming super-aggregates, this peak suffers a hypsochromic shift, which is detected at 322 nm [28,30]. In Figure 2, representative UV/Vis spectra are shown for each of the supramolecular forms of AmB.

#### 2.1.2. Circular Dichroism

CD is a technique that involves the application of circularly polarized light through a sample and the measurement of the absorption difference between right- and left-handed polarized lights [32]. Monomeric AmB presents a spectrum of very low intensity. On the contrary, oligomers present two maxima at 330 and 350 nm, with a dichroic couplet centered at 340 nm. Regarding aggregates, they present the same maxima wavelength when formed through the increase in concentration. Nonetheless, when aggregates are formed through heating, there is a blue shift (close to what is seen in the UV/Vis absorption spectrum), with two maxima at 320 and 340 nm and a dichroic couplet signal centered at 330 nm [30,33,34,35]. The higher the extent of AmB aggregation, the higher the CD intensity (Figure 3).

#### 2.1.3. Fluorescence Techniques

Analyses of steady-state and time-resolved fluorescence, fluorescence anisotropy and fluorescence correlation spectroscopy of AmB (autofluorescence) in different solutions [36,37,38] revealed the formation of dimeric and aggregated species of the drug, even in alkaline solution at pH 12. The fact that these self-associated species appeared when drug molecules were electrically charged (at pH 12) also implied an antiparallel orientation of neighboring molecules in the structures [37].

#### 2.1.4. Dynamic Light Scattering for Particle Size Analysis

As expected, the higher the aggregation, the higher the size of the AmB nanoassembly. While an AmB monomer has an average size of 1 nm, oligomers have a size around 40 nm and aggregates around 300 nm [28,39].

#### 2.1.5. DSC and PXRD for Crystallinity Analysis

AmB can be present in the amorphous phase as a monomer or oligomers. Nonetheless, if AmB is in aggregated forms, it has some degree of crystallinity that can be detected by techniques such as DSC and PXRD [28].

### 2.2. AmB Mechanism of Action

The most prominent mechanism of action of AmB involves its binding to ergosterol, the main component of fungal or parasitic cell membranes. AmB tetramers and octamers arrange themselves in cylindric-like structures, with the hydrophobic part of AmB molecules interacting with ergosterol in the lipid bilayer, resulting in multimeric transmembrane pores. The positively charged amino group of AmB is required for its activity, as well as the polyene subunit [12,38,40,41] (Figure 4).

The formation of pores allows the extravasation of electrolytes (such as K^+^, NH_4_^+^ and H_2_PO_4_^−^) from the intracellular environment, in addition to carbohydrates and proteins [23]. Moreover, there is a subsequent influx of protons into the pathogenic cell that causes acidification of the intracellular medium, with precipitation in the cytoplasm and, eventually, cell death [12,42].

It has also been proposed that AmB may adsorb or sequester ergosterol on the membrane surface, destabilizing the phospholipid bilayer and impairing fundamental cellular processes [18,43]. Moreover, AmB can induce oxidative stress, either directly producing or causing intracellular accumulation of reactive oxygen species (ROS) and reactive nitrogen species [44,45]. The oxidation of unsaturated fatty acids of the cell membrane leads to a change in the integrity of the membrane that becomes susceptible to the osmotic shock derived from the formation of transmembrane channels [31].

The supramolecular organization of AmB also exerts influence on its mechanism of action. Data from in vitro assays indicated that AmB activity was higher for the monomeric form and lower for the aggregates, while an intermediate activity was observed for the oligomers. The reason for that can be related to several factors. It was found that aggregated AmB produced less ROS [28,46]. Moreover, the smaller size of monomeric AmB may result in a higher drug penetration, compared to the larger oligomeric and aggregated forms [28,46].

### 2.3. AmB Mechanism of Toxicity

Besides being toxic to fungal and *Leishmania* cells, AmB can be toxic to human tissues. AmB toxicity is related to its binding to cholesterol, oxidation of cell membranes and production of ROS. Nephrotoxicity represents an AmB major side effect, with AmB supramolecular organization influencing this effect.

It is well known that the size of the drug molecules and their supramolecular species interfere in their biodistribution. It has been demonstrated that while particles with an average diameter larger than 20 nm did not accumulate significantly in the kidneys, those with less than 5 nm were effectively retained therein [47]. One third of AmB total clearance is renal [12]; therefore, nephrotoxicity occurs when AmB passes through the kidneys to be eliminated in urine. As monomeric AmB has a very small size (around 1 nm), it is preferentially eliminated in the urine. On the contrary, as AmB aggregates have a higher average size (>300 nm), they tend to accumulate more in the liver and spleen and are less nephrotoxic [28,39].

Moreover, the interaction of AmB oligomers with cholesterol was found to favor the formation of AmB tetramers and octamers, leading to cell death. In contrast, AmB monomers’ interaction with cholesterol did not lead to oligomer formation. In that sense, it was documented that AmB monomers were safer than AmB oligomers in vivo [48]. From this data, it has been suggested that AmB monomers might be a safer option compared to oligomers [18,42].

Finally, AmB aggregates may act as AmB reservoirs, releasing monomers over time. Therefore, they can be seen as a more specific option than AmB oligomers, with a similar safety profile compared to the monomers [49,50]. Some studies have also stressed that AmB aggregates may be even more specific and safer than AmB monomers [28,51].

## 3. Basic Characteristics of Liposomes

Liposomes have been extensively studied for their interaction with AmB, to investigate both the drug mechanisms of action and toxicity and to explore their potential as drug carrier systems.

Liposomes are enclosed vesicles of concentric lipid bilayer, generally made from phospholipids, such as the zwitterionic phosphatidylcholine (PC) [52]. Membranes made from phospholipids with a phase transition temperature (Tt) higher than 37 °C (e.g., dipalmitoylphosphatidylcholine or DPPC) are in a rigid gel state in physiological conditions, whereas membranes made from phospholipids with a transition temperature lower than 37 °C (e.g., dimyristoylphosphatidylcholine or DMPC and PC from a natural source) are in the fluid liquid-crystalline state. The inclusion of cholesterol in the PC membrane in a proportion higher than 30 mol% results in the disappearance of the membrane phase transition and maintains membrane fluidity in an intermediate state between gel and liquid crystalline. Rigid liposomes and sterol-containing liposomes show improved stability in biological fluids, such as serum [53] and intestinal fluid [54]. Electrically charged lipids, such as the negatively charged phosphatidylglycerol (PG) or positively charged stearylamine (SA), can be included in the membrane to improve the colloidal stability of the vesicle suspension and enhance the surface binding of oppositely charged substances. Liposomes, like other colloidal nanosystems, are rapidly cleared from the blood circulation by macrophages of the mononuclear phagocyte system (MPS), allowing passive drug targeting to the liver, spleen and bone marrow [53]. The recognition of liposomes by macrophages is facilitated by opsonization upon binding of serum components onto the vesicle surface. Lipids with polar head groups consisting of long-chain polyethylene glycol (PEG, typically with 2000 or 5000 kDa) can also be added in a proportion of 5 to 10 mol%, allowing the surface coating with a hydrophilic layer, improving the colloidal stability of the vesicles, reducing the cellular uptake and prolonging the liposome blood circulation [55].

The coating of the liposome surface with layers of polymers, such as PEG; enteric polymers; proteins; or chitosan was found to be effective in protecting the vesicles and their drug content from the harsh gastrointestinal environment and improving oral drug delivery [56,57,58].

Lipid vesicles have also been used in topical formulations to carry drugs and enhance their penetration through the skin. Conventional PC liposomes generally accumulate in the *stratum corneum*, upper skin layers and the appendages, and they show minimal penetration to deeper tissues [59,60]. The deformability of the liposomes can be manipulated through incorporation of an edge activator, such as sodium cholate or Tween 80, in the phospholipid membrane [61,62]. The resulting liposomes, named ultradeformable, flexible or elastic vesicles, were claimed to improve the penetration of a range of small hydrophilic and lipophilic molecules, peptides and macromolecules into deep peripheral tissues and/or into the systemic blood circulation. Figure 5 illustrates the relationship between the lipid composition, the membrane physical state and functional properties of liposomes.

## 4. Reduced Toxicity of Liposomal AmB Formulations: Role of the Aggregation State of AmB and the Rate of Drug Release

A major benefit of liposomes compared to other drug nanocarriers is the marked reduction of AmB toxicity [9]. Liposomes also distinguish themselves among drug nanocarriers for their elevated biocompatibility and high versatility. Those can be prepared from different sizes, lamellarities and lipid compositions, allowing modulation of membrane fluidity, membrane surface characteristics, vesicle deformability and, ultimately, membrane–drug interaction.

Early studies have indicated that the size and lipid composition of liposomal AmB strongly affect the in vivo toxicity of the formulation [63,64,65]. The rank order of reduction of lethality is: sterol-containing liposomes > rigid liposomes > fluid liposomes [64]. The proportion of negatively charged phospholipid (DSPG) was also reported to have influence on the lethality, being less with a DSPG:AmB molar ratio of 2:1 than 1:1 [65]. These studies led to the identification of key physicochemical attributes for AmBisome^®^: small vesicle size (diameter < 100 nm); gel phase membrane made from high Tt phospholipids (hydrogenated soy PC and DSPG); and inclusion of cholesterol and negatively charged phospholipid (DSPG), which both interact with AmB [17,65].

From studies of the kinetic of lethality of different liposomal AmB formulations, Szoka et al. suggested that the rate of transfer of AmB may be critical for toxicity [64]. This model is also supported by the work of Bolard et al. showing that vesicles in the fluid state transferred AmB more rapidly than vesicles in the solid state [66].

Early works on the interaction of AmB with liposomes revealed that the lipid composition significantly affects the type of AmB-lipid complex formed, but no information on the molecular forms of AmB was available at that time [66,67].

Recently, detailed information on the molecular organization, localization and orientation of AmB in lipid bilayers was obtained by electronic absorption, linear and CD, fluorescence spectroscopy and confocal fluorescence lifetime imaging microscopy [38,39,68,69,70]. Four main molecular forms of AmB were identified: monomer, parallel and antiparallel dimers, and tetramers formed out the dimers. These species were fully characterized regarding their energy level diagram, showing absorption peaks at 410 nm (monomer), 388 nm (antiparallel dimer), 340 nm (parallel dimer) and 335 nm (tetramer) [39]. Most of these studies were carried out with DPPC membranes, containing sterol or not. While the monomer and parallel dimer species predominated during the interaction of AmB with the DPPC membrane and were located in the surface region of the polar headgroup zone, the inclusion of sterol in the DPPC membrane promoted anti-parallel dimers, as well as tetramers, which penetrated within the membrane hydrophobic core and tended to adopt a vertical orientation with respect to the plane of the lipid bilayer [39,70]. It was also found that the proportion of tetramers increased with the drug/lipid ratio [39]. Comparison of the DPPC membrane in the fluid and gel phases showed that the fluid state favored the transformation of dimers into monomer and the drug penetration into the hydrophobic core of the membrane [69]. Thus, these studies taken altogether indicate that the lipid composition, the membrane fluidity and the drug–to–lipid ratio exert a marked influence on the AmB aggregation profile. Figure 6 illustrates the molecular species identified for their interaction in three different physical states of the membrane.

These structural studies contributed nicely to a better understanding of the mode of interaction of AmB with biological membranes and its mechanisms of action and toxicity. However, extrapolation of this data to the molecular organization of AmB in commercial liposome formulations is not straightforward because these formulations usually contain a high proportion of DSPG, which may form other complexes with AmB [71]. Thus, the ultimate molecular organization of AmB in commercial liposomal AmB remains to be elucidated.

As an emerging concept, it was found that not only the lipid composition, but also the process of preparation of liposomal AmB formulation can affect the AmB aggregation state and the final toxicity of the formulation. This hypothesis gained support after the observation that liposomal AmB formulations Anfogen^®^ and Lambin^®^ were 5–10-fold more toxic than AmBisome^®^, despite their identical chemical compositions [17]. Recently, Rivnay et al. [72] studied liposomal AmB formulations prepared from the same lipid composition as AmBisome^®^ (hydrogenated soy PC, DSPG, cholesterol at 5:2:2.5 molar ratio) and using the process disclosed in the AmBisome^®^ patents [73,74]. The process is based first on the formation of a drug–DSPG complex in a methanol–chloroform mixture, followed by spray drying, hydration and liposome formation by microfluidization, and lyophilization. The authors reported that extended heat treatment of a liposomal AmB suspension just after microfluidization at a temperature above that of the lipid phase transition temperature (65 °C) brought marked changes in the aggregated state of the drug in the liposome bilayer, as followed by changes in the UV/Vis spectrum. The position of the main absorption peak shifted from 335 nm to 321 nm (the same value of AmBisome^®^), and the ratio of the monomer to main peak decreased. This spectral shift was attributed to AmB superaggregation. Importantly, such a “heat curing” process resulted in a 5–10-fold reduction in the in vitro toxicity of the drug product, bringing it close to the values for AmBisome^®^, as measured by the red blood cells (RBC) potassium release assay.

The recent development of a method for measuring AmB release from liposomes under sink conditions [75] led to the demonstration that “heat curing” of liposomal AmB resulted in a reduced rate of AmB release [76]. Thus, the authors proposed that the release rate of AmB is the link between the AmB aggregation state in the liposomal bilayer and the in vitro toxicity of the formulation.

In another recent study, our group has proposed an alternative process for encapsulating AmB into liposomes [77]. This process consists of the incorporation of AmB into preformed empty liposomes, then exploiting the effect of pH on the solubility and aggregation state of AmB and the influence of temperature on membrane fluidity and insertion of AmB into the liposomal membrane. First, AmB was dissolved in NaOH solution. The solution of AmB was then added to a suspension of preformed empty unilamellar vesicles in water. After a short incubation period at 60 °C, the pH was adjusted to neutral value (between 6.5 and 6.8). The resulting suspension was finally heated for 5 min at 60 °C. Interestingly, the UV/Vis absorption and CD spectra of the final formulation differed from those of AmBisome^®^, despite similar lipid compositions (Figure 7). The absorption peak of the new formulation was centered at 328 nm, evidencing a red shift in comparison to AmBisome^®^. The CD spectrum of the new formulation exhibited low-intensity bands, in contrast to AmBisome^®^ that showed an intense couplet-type signal (Figure 7). It was suggested that AmB is less aggregated in the new formulation, when compared to AmBisome^®^. Evaluation of the kinetic of drug release evidenced much faster drug release from the new formulation in comparison to AmBisome^®^ (Figure 8). As illustrated in Figure 7, when our formulation was submitted to “heat curing” for 4 h at 60 °C, the absorption peak maximum shifted to the same value as that of AmBisome^®^, and a couplet-type signal appeared in the CD spectrum, supporting a higher order of aggregation after prolonged heating, in agreement with a previous study [72]. Thus, our work further supports the concept that the process of preparation of a liposomal AmB formulation affects the AmB aggregation state and release rate.

## 5. Injectable Liposomal AmB Formulations: Factors Influencing the Antileishmanial Efficacy

In the clinics, AmBisome^®^ is administered by the intravenous route to treat both VL and CL. Experimentally, different liposomal AmB formulations have been investigated to better understand the influence of lipid composition on the therapeutic efficacy and as an attempt to identify the most effective liposomal formulation. As described above, a major benefit of liposomal encapsulation of AmB is the reduction of drug toxicity. Regarding the drug efficacy, treatment of VL also explores liposome-mediated passive drug targeting of the main infection sites, i.e., the liver, spleen and bone marrow. On the other hand, the treatment of CL has exploited the long circulating properties of some liposomal formulations.

### 5.1. Influence of the Lipid Composition

In a pioneering study, New et al. have investigated the influence of lipid composition on the efficacy of liposomal AmB formulations in a murine model of VL [63]. AmB incorporated in cholesterol-containing PC liposomes promoted a much larger parasite clearance in the liver than did cholesterol-free liposomes. Furthermore, rigid liposomes made from hydrogenated PC were more effective than fluid liposomes in reducing the parasite load. Cholesterol-containing rigid liposomes were at least six times more potent than the free drug. The authors also compared the influence of different sterols and reported that ergosterol-containing liposomes were much more effective than liposomes containing cholesterol. It was suggested that a slower drug release rate from the nanocarrier may favor the drug therapeutic efficacy. Interestingly, an opposite tendency was observed for the drug toxicity, which decreased with the slower drug release. This apparent discrepancy may be attributed to the active participation of the host cell in the processing and delivery of the drug to the parasites.

A new lipid, in which two molecules of stigmasterol (an inexpensive plant sterol) are covalently linked to glycerophosphocholine, was used to prepare a new liposomal AmB formulation. The formulation exhibited good colloidal stability and high maximum-tolerated dose by the intravenous route [78]. It showed serum profile and tissue concentrations of AmB similar to those of AmBisome^®^, after intravenous injection in mice. Its antileishmanial activity was also comparable to that of AmBisome^®^ in *Leishmania major*-infected BALB/c mice [79].

Banerjee et al. reported the incorporation of AmB into liposomes made from PC and SA in the molar ratio of 7:0.9. A 70% drug encapsulation efficiency was reported. Treatment of *L. donovani*-infected mice with a single dose of this formulation at 3.5 mg AmB/kg almost completely eliminated the parasites from the infected liver and spleen and promoted a Th1-biased curative immune response [80]. A reduced level of IL-10 and high levels of IFN-γ and IL-12 were observed after 3 months of treatment. The high efficacy of this formulation was attributed to the combined antileishmanial actions of SA-PC liposomes and AmB, as well as the resulting immunodulatory effect.

Table 1 summarizes the factors identified thus far as affecting the therapeutic efficacy of liposomal AmB in murine models of VL and CL, after parenteral administration.

### 5.2. Influence of Liposome Size and PEGylation for CL

Recently, AmBisome^®^ was compared to another liposomal AmB product marketed in India (Fungisome^®^), regarding antileishmanial efficacy and intralesional drug accumulation, after parenteral administration in *L. major*-infected BALB/c mice [81]. Significantly higher therapeutic efficacy and drug accumulation within the infected lesions were observed after AmBisome^®^. This difference was attributed to the smaller size of the vesicles in AmBisome^®^ in comparison to Fungisome^®^ and the expected longer circulation time of small-sized vesicles in the bloodstream [81]. Thus, smaller vesicles most probably extravasated to a higher extent through the leaky capillaries in the inflamed lesion skin.

More recently, PEGylated liposomal AmB was prepared through incorporation of AmB into pre-formed liposomes containing the same lipids as AmBisome^®^, except for the inclusion of 4.7 mol% PEGylated lipid (DSPE-PEG2000) [77]. The formulation was evaluated for its therapeutic efficacy in *L. amazonensis*-infected BALB/c mice after parenteral administration. The PEGylated formulation significantly inhibited the lesion size growth and reduced the lesion parasite load, in comparison to saline-treated control. Strikingly, AmBisome^®^ given under the same regimen was less effective than PEGylated liposomal AmB in controlling lesion size growth and did not significantly reduce the parasite load. The authors raised two possible explanations for the superiority of the PEGylated formulation in comparison to AmBisome^®^. First, PEGylation may confer long-circulating characteristics to the vesicles and enhance the extravasation of liposomes through the leaky capillaries in the inflamed lesion skin. Second, AmB may be more readily available from the PEGylated formulation than AmBisome^®^. As shown in Figure 8, the PEG formulation exhibited much faster drug release compared to AmBisome^®^, supporting the model of improved availability of AmB. The faster drug release is also consistent with the results of CD analysis, indicating a less aggregated state of AmB in PEGylated liposomes [77]. Thus, this work highlights the great potential of the PEGylated liposomal formulation for the treatment of disseminated infections, comprising not only leishmaniasis, but also life-threatening systemic fungal infections.

## 6. Topical Liposomal Formulations of AmB

### 6.1. Topical Delivery

Topical treatments are especially attractive for uncomplicated CL cases, offering significant advantages over systemic therapy, including easier administration, fewer adverse effects and cost-effectiveness. Despite being a very attractive route of administration, the topical route represents a challenge for many drugs. The skin has a complex architecture, as can be seen in Figure 9, and serves as a natural protection against the external environment, being composed of three main layers: the epidermis, dermis and hypodermis. The outermost layer, the epidermis, is composed of two main layers: stratum corneum (SC) and viable epidermis. The SC, composed of corneocytes embedded in a lipid matrix, represents the main physical barrier of the skin, protecting the inner layers from the external environment. The viable epidermis is composed mainly of keratinocytes, melanocytes, Merkel cells and Langerhans cells. Adjacent to the epidermis is the dermis, which performs important functions of nutrition and support. The innermost layer of the skin, the hypodermis, is a fat layer providing mechanical support and thermal insulation [82,83,84].

After topical application, drugs can permeate through the skin by three different pathways (Figure 9): (i) appendageal or transfollicular, allowing the direct transport of substances via hair follicles and glandular ducts; (ii) intercellular or paracellular pathway, in which the drug diffuses between the cells, passing through the lipid matrix; and (iii) the intracellular or transcellular pathway, in which the drug passes inside the skin cells and through the lipid matrix [84,85,86]. It is assumed that a combination of these three pathways can contribute to the skin penetration of all substances, but the preferred route depends on their physicochemical characteristics [87].

In CL treatment, the goal of a topical formulation would be to target the infected macrophages located in the dermis [88]. An important aspect of CL is that the patient’s skin is not always intact, as, with the evolution of the disease, the SC is usually damaged. In CL, a papule initially forms at the inoculation site, which usually evolves into ulcerated lesions. In this process, there is a loss of the epidermis and part of the dermis as a result of the local inflammatory response [89]. Although this loss of SC initially facilitates the entry of drugs through the skin, re-epithelialization and wound healing during treatment, along with the production of collagen and metalloproteases of the extracellular matrix, may represent an additional challenge for topical treatment [90]. Thus, it is desired that the formulation works in all possible situations: intact, partially or completely damaged skin [91].

**Figure 9 pharmaceutics-15-00099-f009:**
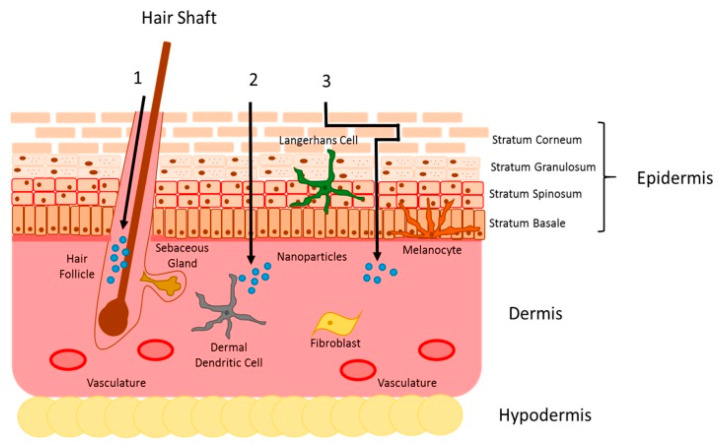
Representation of skin structures and drug penetration pathways. Topically applied drugs can penetrate the skin in one of three different ways: (1) through the appendageal route, (2) through the intracellular route or (3) through the intercellular route. Use authorized by Ref. [92].

To ensure efficient penetration of a substance through the intact skin, it has been proposed that it must have some characteristics, such as a melting point of less than 200 °C, low molecular weight (less than 500 Da) and log *p* value between 1–3 [86]. The ability of the molecule to form hydrogen bonds and its degrees of ionization also need to be considered [83]. Other factors related to the individual, as well as the environment, can also impact skin permeation, such as age, hormonal balance, sebum production, skin hydration and pH gradient [84].

AmB has unfavorable physicochemical characteristics for topical administration, such as high molecular weight (~924 Da), amphoteric nature, low aqueous solubility at physiological pH and tendency to self-aggregation [88,90,93,94]. Its poor permeability across biological barriers severely limits its effectiveness, as reported by López and colleagues [94] in a clinical trial (NCT01845727) using a cream containing 3% AmB (Anfoleish^®^). Although safe, this formulation showed low efficacy in patients with CL, which was attributed to the low transdermal permeation confirmed by the absence of AmB in patients’ plasma [94].

Thus, alternative formulations capable of promoting the topical delivery of AmB are needed. To improve the dermal penetration of drugs and topical therapy, different strategies can be used, such as passive methods (chemical permeation enhancers, for example), physical methods and nanocarriers. The nanocarriers offer a gentler alternative to facilitate drug permeation, being the least damaging one and capable of increasing the drug residence time in the skin [87]. In this sense, recent studies have investigated the use of different types of nanosystems to improve AmB skin permeation, including liposomes, lipid nanoparticles and polymeric carriers [95,96,97,98,99,100,101].

### 6.2. AmB Delivery: Liposomes for Topical Management of CL

Liposomes were the first lipid nanocarriers investigated and marketed to enhance drug penetration into the skin for dermatological and cosmetic applications. The skin delivery of active substances by liposomes is highly affected by their lipid constituents, particle size, surface charge and lamellarity [93]. Over the past three decades, significant progress has been achieved in the design of more deformable vesicles, in particular niosomes, transfersomes and ethosomes, allowing delivery of drugs deeper into/through the skin [82,87]. Several recent studies with liposomal AmB formulations have shown promising results for the topical treatment of CL, which are summarized in Table 2.

Jaafari et al. [101] have investigated liposomes loaded with AmB at different concentrations: 0.1, 0.2 and 0.4% (Lip-AmB). An in vitro permeation study using intact mice skin showed that increasing the AmB concentration in the formulation resulted in a greater amount of AmB permeating the skin. In vivo studies on BALB/c mice infected by *L. major* showed that the efficacy of Lip-AmB 0.4% was greater compared to the other groups (Lip-AmB 0.1 and 0.2%, empty liposomes or PBS). According to the authors, the presence of skin permeation enhancers in liposomes could contribute to these positive results: significant reduction in lesion size and almost complete elimination of parasites in the skin and spleen [101,102]. The results led to development of topical Lip-AmB (0.4%) (Sina Ampholeish^®^) [103].

**Table 2 pharmaceutics-15-00099-t002:** Main experimental studies of AmB-loaded liposomes to treat CL.

Composition	Permeation Studies Outcomes	Animal Model (Dose, Regimen)	In Vivo Outcomes	Reference
AmB; Soy phosphatidylcholine and Tween-80	Higher AmB penetration in SC and in viable epidermis compared to AmBisome^®^ after topical application in intact human skin.	NA *	NA *	[104]
AmB; Soy phosphatidylcholine and sodium cholate	Deeper penetration of AmB and to a larger extent compared to conventional liposomes, after topical application in intact human skin (Franz diffusion cell).	NA *	NA *	[88]
AmB; Soy phosphatidylcholine and Tween-80	Increased drug retention in viable epidermis compared with free AmB, after topical application in intact pig skin (Franz diffusion cell).	NA *	NA *	[105]
AmB deoxycholate and meglumine antimoniate (Glucantime^®^); Span 40; Tween 40; cholesterol; Carbopol^®^ 934 and triethanolamine	NA	BALB/c mice (twice daily for 30 days)	Significant reduction in lesion size after topical treatment with niosomes co-encapsulating AmB and Glucantime^®^ compared to placebo gel (*p* < 0.001) and intramuscular Glucantime^®^ in *L. major*-infected mice. Complete lesion healing not observed.	[95]
AmB and miltefosine; Phospholipon 90G; Tween-80; Carbopol^®^ 934 and triethanolamine	6-fold greater AmB permeation of AmB, compared to AmB simple gel applied topically in intact mouse skin (Franz diffusion cell).	BALB/c mice (AmB 1.5 mg/kg/day twice daily for 4 weeks)	Complete lesion resolution with no signs of scaring in *L. mexicana*-infected mice after topical treatment with co-loaded AmB-miltefosine deformable liposomes. Significant reduction in parasite load at lesion site compared to placebo gel control, AmB gel or single AmB in deformable liposomes.	[96]
AmB; sodium deoxycholate; Soy phosphatidylcholine; ethanol and mannitol	Enhanced permeation across intact mouse skin, compared to previously described liposomal formulations. The in vivo skin pharmacokinetic showed permeability and accumulation within the dermis at therapeutic concentrations for CL treatment.	BALB/c mice (0.5 mg/mL, 20 mg of formulation/day, once a day for 10 consecutive days)	Significant reduction in lesion size compared to the control group (untreated) and almost complete reduction in parasite load at lesion site, after topical treatment in *L. amazonensis*-infected mice.	[97]
AmB; Soy phosphatidylcholine; Cholesterol; Dimethyl sulfoxide; Propylene glycol; Oleic acid; Vitamin E; Methylparaben and Propylparaben	Greater amount of permeated AmB through skin from AmB-liposome (0.4%), compared formulation with lower AmB concentration in permeation study after topical application on intact BALB/c mouse skin (Franz diffusion cell).	BALB/c mice (50 mg liposomal AmB 0.4%, twice a day, for four weeks)	Higher efficacy of liposomal AmB formulation (0.4%) after topical treatment in *L. major*-infected mice, based on the significant reduction (*p* < 0.001) in lesion size and almost complete elimination of parasite load in skin and spleen compared to control groups (PBS or empty liposomes).	[101,103]

* NA means not applicable, when no study was performed for the described parameter.

Other interesting studies in AmB topical delivery have explored the potential of ultra-deformable liposomes (AmB-UDL), using Tween 80^®^, sodium cholate or sodium deoxycholate as an edge activator. Perez et al. [104] noticed that the insertion of AmB reduced vesicle deformability. This finding is in line with other reports in the literature and can be explained by the interaction of AmB with the lipids and edge activators, reducing their mobility [88,104,105,106]. As shown by the authors, the increase in AmB content, in addition to reducing the deformability, modified the absorption spectrum, suggesting AmB self-association in liposome bilayers. An interesting observation was that increasing the surfactant concentration could circumvent this event, keeping AmB in the monomeric form [104]. This probably explains the improved in vitro skin penetration of AmB from this formulation, in comparison to the AmBisome^®^. Carvalheiro et al. [105] conducted studies evaluating liposomes of similar composition, confirming that ultra-deformable liposomes promoted increased drug penetration into the skin. In addition, Peralta et al. [88] also showed that this type of liposome provided better drug penetration into/through human skin than conventional ones. The studies presented above showed improvement in AmB’s topical delivery. However, in vivo proof of concept was not performed. For a more complete view, studies on experimental models of CL are described below.

Fernández-García et al. [97] developed another AmB formulation in ultradeformable vesicles (AmB-TF) and evaluated in vitro the drug permeation across intact mice skin. Although there was no significant difference in permeation between the AmB-TF and AmB-DMSO solutions, the permeation flux from AmB-TF was about five times higher than that described previously for other liposomal formulations, including transferosomes loaded with AmB. The in vivo skin pharmacokinetic of AmB-TF was also assessed after topical administration in mice and showed permeation and accumulation of AmB in the dermis at therapeutic concentrations relevant for the treatment of leishmaniasis. In line with these findings, the topical application of AmB-TF in mice experimentally infected by *L. amazonensis* over 10 days resulted in almost complete elimination of the parasite burden in the lesion, which was similar to that observed after intralesionally administered Glucantime^®^. Regarding the effect on the lesion size, the efficacy of intralesional Glucantime^®^ was greater than that of AmB-TF. However, the overall data suggested that increasing treatment time or twice-daily application of topical AmB-TF could lead to complete lesion healing.

In addition to using ultradeformable vesicles, some authors used another strategy: the drug combination. Mostafavi et al. [95] evaluated niosomes co-encapsulating AmB and Glucantime^®^ (AmB-Glucantime^®^ niosomes), composed by Span 40 and Tween 40, whereas Dar et al. [96] investigated ultradeformable liposomes co-loaded with AmB and miltefosine (AmB-MTF liposomes), composed by PC and Tween 80. Despite the large particle size of the niosomes co-encapsulating AmB and Glucantime^®^ (9.5 μm), the topical treatment of BALB/c mice infected with *L. major* (twice daily for 30 days) promoted reduction in the lesion in comparison to placebo and intramuscular Glucantime^®^ [96]. However, an important gap in this study was the lack of parasite load assessment in the lesion, an important parameter to evaluate formulation effectiveness. On the contrary, the topical treatment with AmB-MTF liposomes developed by Dar et al. [96] resulted in complete lesion resolution in mice infected with *L. mexicana* after twice daily treatment for 4 weeks. In agreement, the lesion parasite burden had a significant reduction for AmB-MTF liposomes when compared to the other groups—untreated, treated with plain AmB-gel or treated with ultradeformable liposome gel containing only AmB [96]. These results confirmed the benefit of drug combination due to a possible synergistic effect in CL treatment.

Although there are few clinical trials investigating AmB topical formulations, Khamesipour et al. [107] investigated the activity of liposomal AmB (0.4%) (SinaAmpholeish^®^) developed by Jaafari et al. [101], which had already shown safety in a Phase I clinical trial [108]. The pilot study compared three treatment groups in patients with CL caused by *L. major*: (i) topical liposomal AmB (0.4%) alone twice daily for 28 days; (ii) topical liposomal AmB (0.4%) in combination with daily intramuscular Glucantime^®^; (iii) weekly intralesional Glucantime^®^ plus biweekly cryotherapy (the standard treatment in the Islamic Republic of Iran). Complete cure was 92%, 95% and 48.5% of patients who received combination treatment (liposomal AmB 0.4% plus Glucantime^®^), topical liposomal AmB only and standard treatment alone (Glucantime^®^ plus cryotherapy), respectively. This study evidenced the great efficacies of liposomal AmB and liposomal AmB in combination with Glucantime^®^.

In turn, Horev et al. [109] performed a randomized, double-blind, placebo-controlled trial to investigate the efficacy of liposomal AmB 0.4% gel in *L. major*-infected patients treated twice daily for 56 days. Different parameters were evaluated, such as lesion diameter, ulceration and healing. At the end of treatment, the results were similar between the liposomal AmB gel-treated and control groups. The authors suggested that a longer treatment duration may be necessary to improve efficacy because clinical improvement, including negative PCR test, was more clearly observed after 56 days rather than earlier.

In the literature there are many reports about the ideal characteristics of the liposome carrier for topical application. However, the effects of the AmB aggregation state on the skin drug penetration and the formulation efficacy are still poorly explored. AmB insertion in lipid vesicles is a complex process because it can adopt different aggregation forms, depending on the AmB concentration, vesicle composition and preparation method [72]. Additionally, the development of new skin models, providing more realistic conditions, is an important point to increase the chance of bringing topical formulations from the bench to the market [84]. In this sense, it is worth noting that all studies presented here performed skin permeation tests on intact skin. This is an important limitation because, under pathological conditions like CL, considerable skin damage usually occurs, altering skin architecture and permeability [91].

## 7. Oral Liposomal Formulations of AmB

The oral route is usually preferred for drug administration. Oral treatments often result in lower drug toxicity in comparison to the parenteral ones and improved patient compliance. This is especially important for neglected tropical diseases, such as leishmaniasis, which affect mainly poor people, who live in remote areas and have limited access to health centers.

However, AmB is a class IV drug, according to the BCS classification system, exhibiting low solubility in neutral pH and low membrane permeability, with expected low oral bioavailability.

Indeed, several physicochemical factors contribute to the low oral bioavailability of AmB from the existing commercial formulations, including AmBisome^®^. These factors comprise the high molecular weight of the AmB molecule, its low solubility in both aqueous and lipidic environments and tendency to self-associate, and its instability at the low pH found in the stomach [110].

This context has stimulated the search for strategies to improve the oral delivery of AmB, with few successful cases [40,110,111,112]. The following drug carriers have shown improvement of AmB bioavailability or efficacy by the oral route: polymeric nanoparticles [113,114], polymer lipid hybrid nanoparticles [115], solid lipid nanoparticles [116], chitosan-coated nanostructured lipid carriers [117], cubosomes [118], emulsions [119,120], cochleates [121,122,123,124] and liposomes [77].

In a recent review, Wasan et al. [111] reported currently investigated AmB formulations for the treatment of parasitic infections, with an emphasis on two oral lipid formulations that have reached clinical trials. First, a self-emulsifying lipid-based formulation (iCo-019), consisting of a mixture of mono- and di-glycerides, in addition to D-alpha-tocopheryl poly(ethylene glycol) succinate, which completed two human Phase I trials. Second, an encochleated AmB deoxycholate formulation under Phase II trials to determine its efficacy for cryptococcal meningitis. Interestingly, a low aggregation state of AmB was claimed for both types of formulations [111,122]. Moreover, the safety, tolerability and pharmacokinetics data of iCo-19 following single doses to healthy humans supported long-lasting systemic drug exposure, with no evidence of gastrointestinal, hepatic or renal toxicities associated with AmB [125]. A similar safety profile has been reported in humans for the encochleated AmB deoxycholate formulation [124]. This first set of clinical data highlights the great potential of these lipid AmB formulations for the oral treatment of leishmaniasis.

Ramos et al. [77] reported for the first time an orally active liposomal AmB formulation. The nanoformulation contained the same lipids as AmBisome^®^, but also included 4.7 mol% DSPE-PEG2000. Characterization of the drug aggregation state by CD suggested lower aggregation of AmB in the PEGylated formulation, when compared to AmBisome^®^. This feature is likely critical, as the liposomal AmB formulation seems to share the low drug aggregation state with oral AmB formulations under clinical trials. As illustrated in Figure 8, the new liposomal AmB formulation exhibited much faster drug release than AmBisome^®^, in agreement with the lower extent of drug aggregation. The PEGylated formulation also showed greater stability in simulated gastric fluid, when compared to the non-PEGylated formulation, regarding particle size distribution and AmB aggregation state. Importantly, the PEGylated liposomal AmB formulation exhibited therapeutic efficacy by the oral route in the murine model of CL, promoting significant inhibition in the lesion size growth and reduction in the parasite load in the lesion, when compared to the saline-treated control. This effect was achieved at a relatively low dose of AmB (5 mg/kg) given on alternate days. The reduced renal toxicity of oral treatment with PEGylated liposomal formulation was also supported by the absence of change in the plasma level of urea, in contrast to AmBisome^®^ given parenterally at the same dosage. Considering the low aggregation state of AmB in the oral liposomal formulation and the significant drug release, one can expect an effective intestinal absorption of AmB under the free form. In this context, a sustained drug release from the liposomal formulation in the intestine may result in a long-lasting drug plasma level and may explain the reduced toxicity, as proposed previously for iCo-019 [125].

The benefit of liposome PEGylation for oral drug delivery is consistent with previous reports in the literature for other drugs, including peptides, proteins and lipophilic substances [58]. As illustrated in Figure 10, the oral efficacy of PEGylated liposomal AmB formulation may be attributed to several factors, including: (i) the prevention of liposome aggregation in an acidic environment, (ii) the protection of liposomes from the action of bile, (iii) the protection of AmB from acidic degradation, and (iv) the low state of AmB aggregation, leading to more effective intestinal drug absorption. Further studies are needed to identify the factors that most contribute to improved oral drug efficacy and further optimize liposomal formulations for the delivery of AmB by the oral route.

## 8. Conclusions

As highlighted in this review, progress has been achieved toward the identification of the factors that affect the toxicity of liposomal AmB formulations and their efficacy against CL and VL. Previous works have shown that the toxicity and efficacy of the nanoformulation can be controlled through manipulation of the lipid composition and membrane surface of liposomes, as well as the process used for drug incorporation. In this context, the AmB aggregation state and the rate of drug release from liposomes have been identified as critical parameters. In recent years, significant advances have also been achieved in the development of effective liposomal AmB formulations for the oral and topical treatment of CL.

A model has emerged, in which AmB formulations prepared from cholesterol-containing rigid liposomes and submitted to “heat curing” exhibit super-aggregated AmB in the membrane and slow drug release, resulting in low toxicity and high efficacy against VL. On the contrary, it is suggested that liposomal AmB formulations containing the drug under a less aggregated form—as, for instance, with fluid liposomes—exhibit faster drug release and promote improved drug bioavailability by topical and oral routes.

Finally, it is important to mention that gaps of knowledge and challenges still exist for development of innovative liposomal AmB formulations. Those identified by the authors as critical in future investigations are displayed in Table 3.

## Figures and Tables

**Figure 1 pharmaceutics-15-00099-f001:**
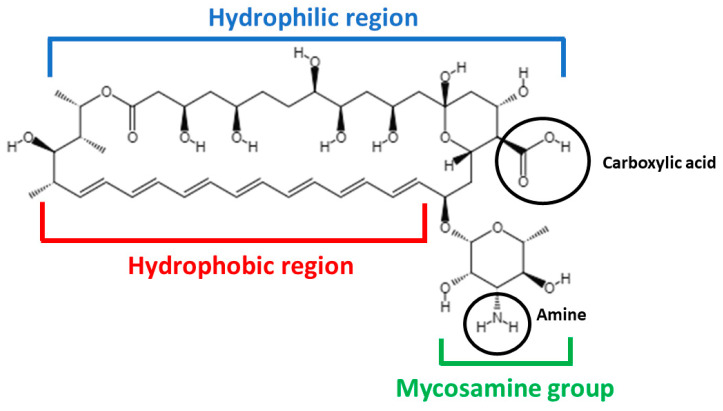
Chemical structure of AmB.

**Figure 2 pharmaceutics-15-00099-f002:**
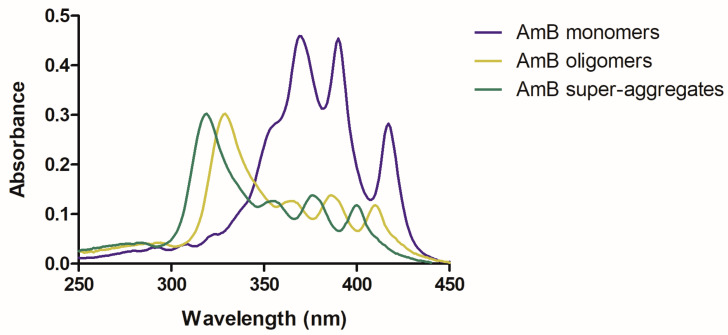
Representative UV/Vis spectra of the different AmB supramolecular forms: monomer (in blue), oligomers (in moss green) and aggregates (in green).

**Figure 3 pharmaceutics-15-00099-f003:**
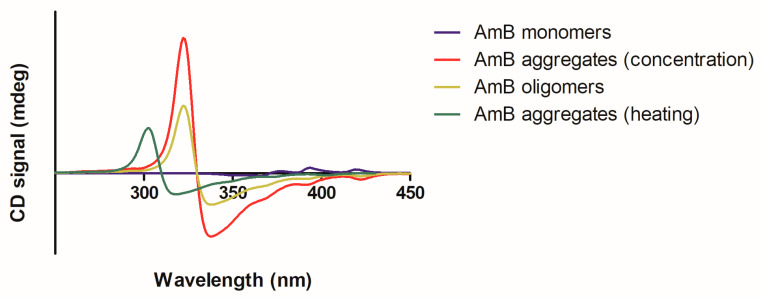
Representative CD spectra of the different AmB supramolecular forms: monomer (in blue); oligomers (moss green); aggregates formed through temperature increase (in green) or concentration increase (in red).

**Figure 4 pharmaceutics-15-00099-f004:**
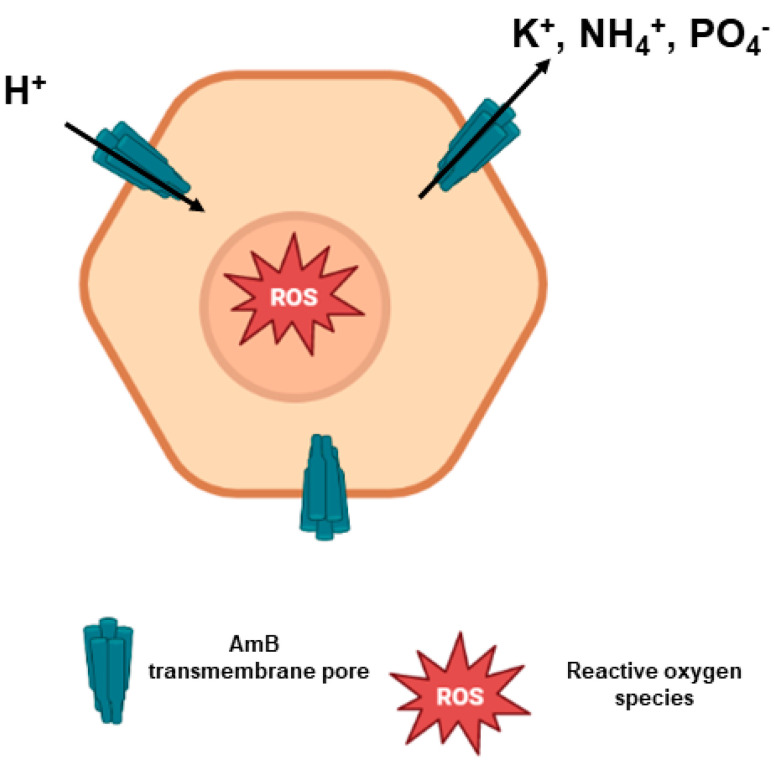
Mode of action of AmB. AmB molecules interact with ergosterol, forming AmB transmembrane pores. There is an intense efflux of ions through these pores that eventually leads to proton influx, causing cell death. Moreover, there is a production of reactive oxygen species (ROS) due to AmB-mediated oxidative stress when the drug crosses the cell membrane.

**Figure 5 pharmaceutics-15-00099-f005:**
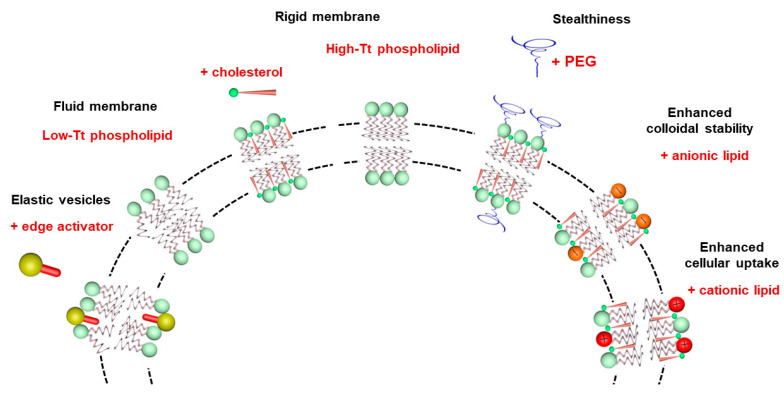
Illustration of the influence of lipid composition on the membrane physical state and functional properties of liposomes.

**Figure 6 pharmaceutics-15-00099-f006:**
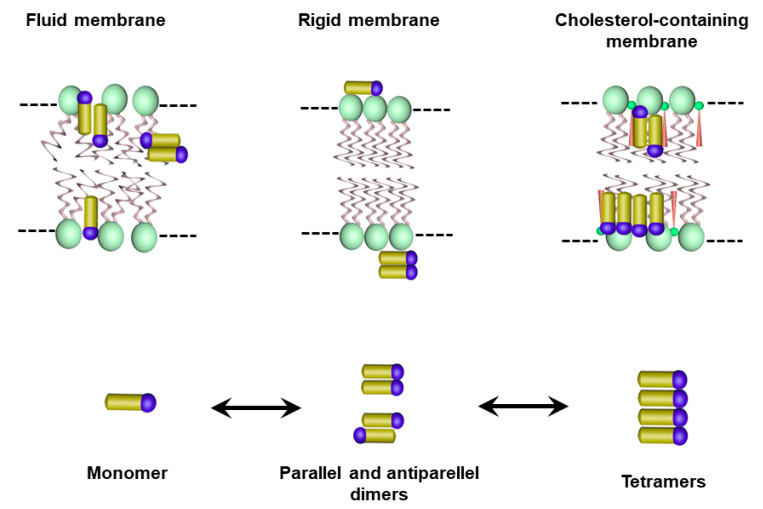
Illustration of the molecular and supramolecular AmB species and their membrane localization, as suggested by previous studies of the interaction of AmB with membranes under different physical states.

**Figure 7 pharmaceutics-15-00099-f007:**
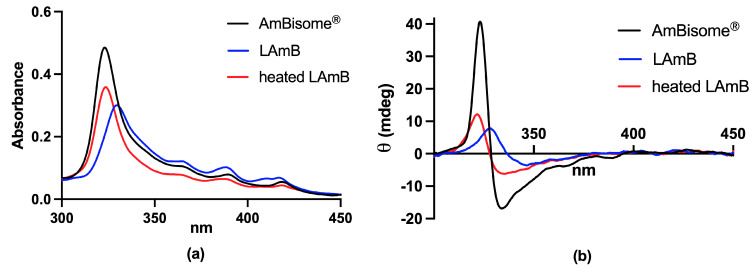
Absorption (**a**) and circular dichroism (**b**) spectra of conventional liposomal AmB (LAmB) prepared according to [77], before and after 4 h heating at 60 °C, in comparison to AmBisome^®^. Lyophilized LAmB was reconstituted with water at 4 mg/mL AmB and submitted or not to heat curing. Spectra were registered at 25 °C after reconstitution of the nanoformulations in water and 100-fold dilution in PBS.

**Figure 8 pharmaceutics-15-00099-f008:**
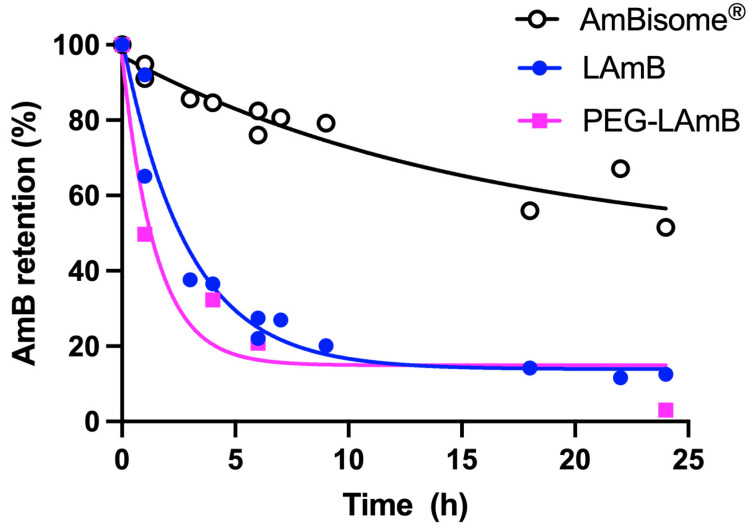
Kinetics of release of AmB at 55 °C from conventional and PEGylated liposomal AmB (LAmB and PEG-LAmB, respectively) prepared according to [77], and AmBisome^®^. Dialysis was performed using Spectra-Por^®^ Float-A-Lyzer^®^ G2 MWCO 100kDa in the presence of 5% γ-cyclodextrin, according to [76].

**Figure 10 pharmaceutics-15-00099-f010:**
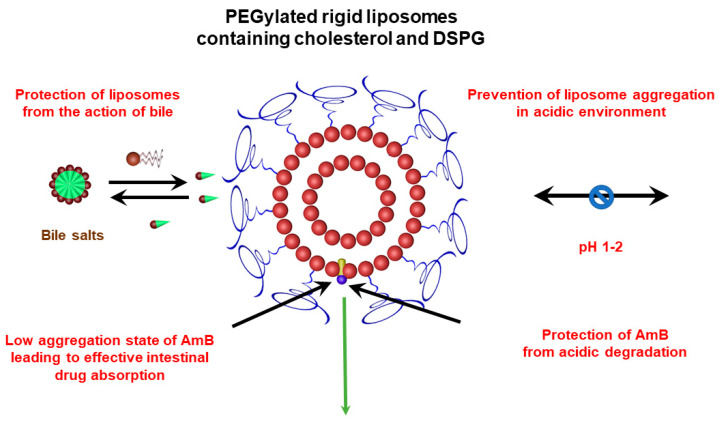
Possible factors contributing to the oral efficacy of PEGylated liposomal AmB formulation.

**Table 1 pharmaceutics-15-00099-t001:** Factors affecting the therapeutic efficacy of liposomal AmB in murine models of VL and CL, after parenteral administration.

Parameter	Observed Therapeutic Effect	Proposed Mechanism	Reference
	Model of visceral leishmaniasis		
Membrane fluidity	Rigid lipo. > fluid lipo.	Slower drug release from rigid liposomes	[63]
Inclusion of cholesterol	Lipo. with cholesterol > Lipo. without cholesterol	Higher affinity of the drug for cholesterol-containing membrane	[63]
Inclusion of ergosterol	Lipo. with ergosterol > Lipo. with cholesterol	Higher affinity of the drug for ergosterol-containing membrane	[63]
	Model of cutaneous leishmaniasis		
Liposome size	Small-sized lipo. > large-sized lipo.	Extended blood circulation of smaller liposomes and higher accumulation in the lesion	[81]
PEGylation of liposomes	PEGylated lipo. > conventional lipo.	Extended blood circulation time of PEGylated liposomes and higher accumulation in the lesion	[77]
Inclusion of stearylamine (SA)	SA-containing lipo. > SA-free lipo.	Combined leishmanicidal and immunodulatory actions	[80]

**Table 3 pharmaceutics-15-00099-t003:** Gaps of knowledge or challenges to be addressed in future investigations of liposomal AmB formulations.

Gap of Knowledge or Challenge
1. Impact of DSPG on the supramolecular organization of AmB in liposomal formulation.
2. Systematic study of the aggregation state of AmB in topical liposomal formulations.
3. Mechanisms responsible for the improved oral efficacy of PEGylated liposomes.
4. Improving the shelf-life stability of liposomal AmB formulations, regarding the effect of temperature.
5. Developing more effective and safe strategies combining the same liposomal system AmB and an immunomodulator.
6. Oral efficacy of PEGylated liposomal AmB in visceral leishmaniasis.
7. Optimizing liposomal AmB for oral delivery and exploring alternative lipid compositions and coating strategies of liposomes.
8. Translating experimental findings into human applications, scaling-up the production and overcoming regulatory issues and the “Neglected Tropical Disease” barrier.

## Data Availability

Not applicable.

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
