# Peer review of "Liposomal Amphotericin B for Treatment of Leishmaniasis: From the Identification of Critical Physicochemical Attributes to the Design of Effective Topical and Oral Formulations"

_pharmaceutics, 2022, doi:10.3390/pharmaceutics15010099_

Round 1

Reviewer 1 Report

Frézard et al. have written a review paper (it appears to have been changed from an article to a review paper) discussing the progress achieved so far toward the understanding of the mechanism responsible for the reduced toxicity of liposomal AmB formulations and the factors that influence their efficacy against leishmaniasis. It also presents the recent advances in the development of more effective liposomal AmB formulations, including topical and oral liposome formulations.

Strengths: This is a well written paper that provides a solid (but not complete) overview of about the current amphotericin B formulations and the accepted mechanism of action on how amphotericin B works. The topic formulation section is well done. 

Limitations: There have been many review articles on amphotericin B for the treatment of leishmaniasis that have been published over the years. Unfortunately, this paper really does not add anything new to the literature. The journal "pharmaceutics" recently just published a paper on October 28th which discusses oral amphotericin B formulations for parasitic infections, such as leishmaniasis which covers many aspects highlighted in this paper (not all). There are many review papers on the use of liposomal amphotericin B for treating VL over the years and a number of new papers published recently.  

The mechanism by which liposomal amphotericin B reduces toxicity has been well described and has been known for a very long time and there really is not anything new here.

Some examples below: 

Stoodley R, Shepherd J, Wasan KM, Bizzotto D. Amphotericin B interactions with a DOPC monolayer. Electrochemical investigations. Biochim Biophys Acta. 2002 Aug 19;1564(1):289-97. doi: 10.1016/s0005-2736(02)00463-7. PMID: 12101024.

Hartsel SC, Baas B, Bauer E, Foree LT Jr, Kindt K, Preis H, Scott A, Kwong EH, Ramaswamy M, Wasan KM. Heat-induced superaggregation of amphotericin B modifies its interaction with serum proteins and lipoproteins and stimulation of TNF-alpha. J Pharm Sci. 2001 Feb;90(2):124-33. doi: 10.1002/1520-6017(200102)90:2<124::aid-jps3>3.0.co;2-x. PMID: 11169529.

Bekersky I, Fielding RM, Dressler DE, Lee JW, Buell DN, Walsh TJ. Plasma protein binding of amphotericin B and pharmacokinetics of bound versus unbound amphotericin B after administration of intravenous liposomal amphotericin B (AmBisome) and amphotericin B deoxycholate. Antimicrob Agents Chemother. 2002 Mar;46(3):834-40. doi: 10.1128/AAC.46.3.834-840.2002. PMID: 11850269; PMCID: PMC127463.

Wasan KM, Lopez-Berestein G. The interaction of liposomal amphotericin B and serum lipoproteins within the biological milieu. J Drug Target. 1994;2(5):373-80. doi: 10.3109/10611869408996812. PMID: 7704481.

Janknegt R, de Marie S, Bakker-Woudenberg IA, Crommelin DJ. Liposomal and lipid formulations of amphotericin B. Clinical pharmacokinetics. Clin Pharmacokinet. 1992 Oct;23(4):279-91. doi: 10.2165/00003088-199223040-00004. PMID: 1395361.

Adler-Moore J, Proffitt RT. AmBisome: liposomal formulation, structure, mechanism of action and pre-clinical experience. J Antimicrob Chemother. 2002 Feb;49 Suppl 1:21-30. doi: 10.1093/jac/49.suppl_1.21. PMID: 11801577.

Laniado-Laborín R, Cabrales-Vargas MN. Amphotericin B: side effects and toxicity. Rev Iberoam Micol. 2009 Dec 31;26(4):223-7. doi: 10.1016/j.riam.2009.06.003. PMID: 19836985.

Mohamed-Ahmed AH, Brocchini S, Croft SL. Recent advances in development of amphotericin B formulations for the treatment of visceral leishmaniasis. Curr Opin Infect Dis. 2012 Dec;25(6):695-702. doi: 10.1097/QCO.0b013e328359eff2. PMID: 23147810.

The oral formulation section is not well done. They do cite a number of papers but do not do a thorough review of the different oral technologies. A table would be helpful. Unfortunately a new paper has just been published that describes this area very well: 

Wasan E, Mandava T, Crespo-Moran P, Nagy A, Wasan KM. Review of Novel Oral Amphotericin B Formulations for the Treatment of Parasitic Infections. Pharmaceutics. 2022 Oct 28;14(11):2316. doi: 10.3390/pharmaceutics14112316. PMID: 36365135.

Wasan KM. Development of an Oral Amphotericin B Formulation as an Alternative Approach to Parenteral Amphotericin B Administration in the Treatment of Blood-Borne Fungal Infections. Curr Pharm Des. 2020;26(14):1521-1523. doi: 10.2174/1381612826666200311130812. PMID: 32160842.

This paper is really not a review article as it highlights the Oral efficacy of PEGylated liposomal AmB in visceral leishmaniasis specifically without doing a comparison to other oral formulations that have reached and completed clinical trials. 

Author Response

Frézard et al. have written a review paper (it appears to have been changed from an article to a review paper) discussing the progress achieved so far toward the understanding of the mechanism responsible for the reduced toxicity of liposomal AmB formulations and the factors that influence their efficacy against leishmaniasis. It also presents the recent advances in the development of more effective liposomal AmB formulations, including topical and oral liposome formulations.

Strengths: This is a well written paper that provides a solid (but not complete) overview of about the current amphotericin B formulations and the accepted mechanism of action on how amphotericin B works. The topic formulation section is well done. 

Authors’ response: The authors thank the Reviewer for his comments on the strengths of our manuscript.

Limitations: There have been many review articles on amphotericin B for the treatment of leishmaniasis that have been published over the years. Unfortunately, this paper really does not add anything new to the literature. The journal "pharmaceutics" recently just published a paper on October 28th which discusses oral amphotericin B formulations for parasitic infections, such as leishmaniasis which covers many aspects highlighted in this paper (not all). There are many review papers on the use of liposomal amphotericin B for treating VL over the years and a number of new papers published recently.

Authors responses:

The review paper mentioned by the Reviewer has been published at about the same time as the present review was submitted. We are sorry for missing it but, fortunately, this excellent review can now be cited in the revised version (please, see the text added in section 7).

It is the opinion of the authors that both Reviews are complementary in their scope and approach.

The published review paper covers amphotericin B formulations used for treatment parasitic infections including leishmaniasis, with an emphasis on oral formulations. However, it does not present an orally effective liposomal amphotericin B formulation and its efficacy against cutaneous leishmaniasis (Ramos et al. Pharmaceutics 2022, 14 (5), 989. https://doi.org/10.3390/pharmaceutics14050989).

Our review focuses on liposomal formulations of amphotericin B for treatment of all forms of leishmaniasis (not only visceral leishmaniasis) and addresses in depth the physicochemical mechanisms involved in their toxicities and efficacies, mainly the role of the drug aggregation state and release rate. We could not find any review paper in the literature with a similar approach. It is also noteworthy that the published review (mentioned by the Reviewer) does not discuss the importance of AmB aggregation state and drug release rate, as the current one.

The mechanism by which liposomal amphotericin B reduces toxicity has been well described and has been known for a very long time and there really is not anything new here.

Some examples below: 

Stoodley R, Shepherd J, Wasan KM, Bizzotto D. Amphotericin B interactions with a DOPC monolayer. Electrochemical investigations. Biochim Biophys Acta. 2002 Aug 19;1564(1):289-97. doi: 10.1016/s0005-2736(02)00463-7. PMID: 12101024.

Hartsel SC, Baas B, Bauer E, Foree LT Jr, Kindt K, Preis H, Scott A, Kwong EH, Ramaswamy M, Wasan KM. Heat-induced superaggregation of amphotericin B modifies its interaction with serum proteins and lipoproteins and stimulation of TNF-alpha. J Pharm Sci. 2001 Feb;90(2):124-33. doi: 10.1002/1520-6017(200102)90:2<124::aid-jps3>3.0.co;2-x. PMID: 11169529.

Bekersky I, Fielding RM, Dressler DE, Lee JW, Buell DN, Walsh TJ. Plasma protein binding of amphotericin B and pharmacokinetics of bound versus unbound amphotericin B after administration of intravenous liposomal amphotericin B (AmBisome) and amphotericin B deoxycholate. Antimicrob Agents Chemother. 2002 Mar;46(3):834-40. doi: 10.1128/AAC.46.3.834-840.2002. PMID: 11850269; PMCID: PMC127463.

Wasan KM, Lopez-Berestein G. The interaction of liposomal amphotericin B and serum lipoproteins within the biological milieu. J Drug Target. 1994;2(5):373-80. doi: 10.3109/10611869408996812. PMID: 7704481.

Janknegt R, de Marie S, Bakker-Woudenberg IA, Crommelin DJ. Liposomal and lipid formulations of amphotericin B. Clinical pharmacokinetics. Clin Pharmacokinet. 1992 Oct;23(4):279-91. doi: 10.2165/00003088-199223040-00004. PMID: 1395361.

Adler-Moore J, Proffitt RT. AmBisome: liposomal formulation, structure, mechanism of action and pre-clinical experience. J Antimicrob Chemother. 2002 Feb;49 Suppl 1:21-30. doi: 10.1093/jac/49.suppl_1.21. PMID: 11801577.

Laniado-Laborín R, Cabrales-Vargas MN. Amphotericin B: side effects and toxicity. Rev Iberoam Micol. 2009 Dec 31;26(4):223-7. doi: 10.1016/j.riam.2009.06.003. PMID: 19836985.

Mohamed-Ahmed AH, Brocchini S, Croft SL. Recent advances in development of amphotericin B formulations for the treatment of visceral leishmaniasis. Curr Opin Infect Dis. 2012 Dec;25(6):695-702. doi: 10.1097/QCO.0b013e328359eff2. PMID: 23147810.

Authors’ responses:

The authors cannot agree with the Reviewer that “there is not anything new” regarding “the mechanism by which liposomal amphotericin B reduces toxicity.”

The relationship between the drug release rate and toxicity has been known for a long time. We preferred to cite the references of pioneer works (Bolard et al. Biochim. Biophys. Acta - Biomembr. 1981, 647, 241–248. https://doi.org/10.1016/0005-2736(81)90252-2; New et al. J Antimicrob Chemother. 1981, 8 (5), 371–381. https://doi.org/10.1093/jac/8.5.371; Witzke et al. Biochemistry 1984, 23, 1668–1674. https://doi.org/10.1021/bi00303a014; Szoka et al. Antimicrob Agents Chemother. 1987, 31 (3), 421–429. https://doi.org/10.1128/aac.31.3.421), rather than every subsequent work that confirmed this relationship.

However, the relationship between the aggregation state of amphotericin B in liposomal membrane and the drug release rate started to be elucidated only recently, from both the study of physicochemical state of AmB in liposome membranes and the direct measurement of AmB release from liposomes.

Please, consider the following recent papers cited in our Review that contribute to our understanding of the physicochemical state of AmB in liposomal membrane and influence of membrane composition (presence of cholesterol; high vs. low transition temperature phospholipids).

  1. Wasko, P.; Luchowski, R.; Tutaj, K.; Grudzinski, W.; Adamkiewicz, P.; Gruszecki, W.I. Toward understanding of toxic side effects of a polyene antibiotic amphotericin B: fluorescence spectroscopy reveals widespread formation of the specific supramolecular structures of the drug. Mol Pharm.2012, 9(5), 1511–1520. https://doi.org/10.1021/mp300143n
  2. Starzyk, J.; Gruszecki, M.; Tutaj, K.; Luchowski, R.; Szlazak, R.; Wasko, P.; Grudzinski, W.; Czub, J.; Gruszecki, W.I. Self-association of amphotericin B: spontaneous formation of molecular structures responsible for the toxic side effects of the antibiotic. J Phys Chem B 2014, 118(48), 13821–13832. https://doi.org/10.1021/jp510245n
  3. Zielińska, J.; Wieczór, M.; Bączek, T.; Gruszecki, M.; Czub, J. Thermodynamics and kinetics of amphotericin B self-association in aqueous solution characterized in molecular detail. Sci Rep. 2016, 6 (1), 19109. https://doi.org/ https://doi.org/10.1038/srep19109
  4. Kamiński, D. M. Recent progress in the study of the interactions of amphotericin B with cholesterol and ergosterol in lipid environments. Eur Biophys J. 2014, 43 (10-11), 453–467. https://doi.org/10.1007/s00249-014-0983-8
  5. Grudzinski, W.; Sagan, J.; Welc, R.; Luchowski, R.; Gruszecki, W.I. Molecular organization, localization and orientation of antifungal antibiotic amphotericin B in a single lipid bilayer. Sci Rep. 2016, 6, 32780. https://doi.org/10.1038/srep32780

Please, also consider the recent papers cited in our Review that bring important insight into the relationship between drug aggregation state, release rate and toxicity.

  1. Rivnay, B.; Wakim, J.; Avery, K.; Petrochenko, P.; Myung, J.H.; Kozak, D.; Yoon, S.; Landrau, N.; Nivorozhkin, A. Critical process parameters in manufacturing of liposomal formulations of amphotericin B. Int J Pharm. 2019, 565, 447–457. https://doi.org/10.1016/j.ijpharm.2019.04.052
  2. Liu, Y.; Mei, Z.; Mei, L.; Tang, J.; Yuan, W.; Srinivasan, S.; Ackermann, R.; Schwendeman, A.S. Analytical method development and comparability study for AmBisome® and generic amphotericin B liposomal products. Eur J Pharm Biopharm. 2020, 157, 241–249. https://doi.org/10.1016/j.ejpb.2020.09.008
  3. Ramos, G.S.; Vallejos, V.M.R.; Borges, G.S.M.; Almeida, R.M.; Alves, I.M.; Aguiar, M.M.G.; Fernandes, C.; Guimarães, P.P.G.; Fujiwara, R.T.; Loiseau, P.M.; et al. Formulation of amphotericin B in PEGylated liposomes for improved treatment of cutaneous leishmaniasis by parenteral and oral routes. Pharmaceutics 2022, 14 (5), 989. https://doi.org/10.3390/pharmaceutics14050989

The oral formulation section is not well done. They do cite a number of papers but do not do a thorough review of the different oral technologies. A table would be helpful. Unfortunately a new paper has just been published that describes this area very well: 

Wasan E, Mandava T, Crespo-Moran P, Nagy A, Wasan KM. Review of Novel Oral Amphotericin B Formulations for the Treatment of Parasitic Infections. Pharmaceutics. 2022 Oct 28;14(11):2316. doi: 10.3390/pharmaceutics14112316. PMID: 36365135.

Wasan KM. Development of an Oral Amphotericin B Formulation as an Alternative Approach to Parenteral Amphotericin B Administration in the Treatment of Blood-Borne Fungal Infections. Curr Pharm Des. 2020;26(14):1521-1523. doi: 10.2174/1381612826666200311130812. PMID: 32160842.

Authors’ responses:

The aim of our manuscript is to review different liposome-based strategies for delivery of amphotericin B, including the oral route. A thorough review of the different oral technologies is also interesting, but it is out of the scope of our review and, as mentioned by the Reviewer, this was already well done in recently published review papers. Thus, it would not be appropriate to reproduce a Table of the different oral technologies, as already reported in these Reviews. To address the Reviewer concerns, the recent review papers mentioned by the Reviewer are now cited and commented in section 7 of our manuscript.

In accordance with the scope of our Review, the “oral formulation section” focused on the unique liposomal formulation of amphotericin B that showed efficacy against leishmaniasis. This is an important contribution of our Review, as this work (Ramos et al. Pharmaceutics 2022, 14 (5), 989. https://doi.org/10.3390/pharmaceutics14050989) was not presented in previous review papers. This section also discusses the possible role of amphotericin B aggregation state and release rate in oral efficacy, which is relevant to other oral technologies.

This paper is really not a review article as it highlights the Oral efficacy of PEGylated liposomal AmB in visceral leishmaniasis specifically without doing a comparison to other oral formulations that have reached and completed clinical trials. 

Authors’ responses:

We agree with the Reviewer and thank for the criticism. Thus, comparison is now presented to other oral formulations that have reached and completed clinical trials.

Therefore, the most advanced oral formulations of AmB are now commented in section 7, together with the results of Phase I clinical trials. We also point in the revised manuscript that “this first set of clinical data highlights the great potential of these lipid AmB formulations for the oral treatment of leishmaniasis.”  

Following the description of the results obtained with liposomal formulations of AmB by oral route, the following paragraph was added to section 7:  

“Considering the low aggregation state of AmB in the oral liposomal formulation and the significant drug release, one would expect a more effective intestinal absorption of AmB under the free form. In this context, a sustained drug release from the liposomal formulation in the intestine may result in long-lasting drug plasma level and may explain the reduced toxicity, as proposed previously for iCo-019 [126].”

Comparison is focused on the lower aggregation state of AmB in these formulations and the reduction of toxicity. Additional comparisons are still limited since the oral efficacy of the PEGylated liposomal AmB was demonstrated in cutaneous leishmaniasis and other oral formulations of AmB were tested only against visceral leishmaniasis. Moreover, there no pharmacokinetic data yet for the liposomal AmB formulation.  

The authors thank the Reviewer for his valuable comments that help to improve greatly our manuscript.

Reviewer 2 Report

This is a review article summarising different liposomal formulations containing amphotericin B. Overall, the review article highlights different challenges in amphotericin B formulation, such as, alternatives to the intravenous route, but also describes the physicochemical and biological behaviour of the drug. Apart from that, a nice description of the skin physiology is described in section 6.1. I believe the literature review carried out is extensive, as many different topics are covered through the manuscript, proving that the authors have a wide knowledge in the drug physicochemistry and an excellent understanding of pharmaceutical technology.

The manuscript is well written in terms of content and structure, however, I would suggest the authors to read carefully through it as some grammar mistakes have been found (e.g. was instead or were, and vice versa), be specially careful with the verb tenses and if the subject is singular or plural. A few errors have been found (e.g. phosphadidylcholine in section 3). Also, the figures are clear.

I think this manuscript is highly valuable and suitable for publication in Pharmaceutics, but a careful revision needs to be done before being accepted for publication.

Author Response

Authors’ responses:

As recommended by the Reviewer, a careful revision of the manuscript was carried out, mainly regarding the English grammar.

The authors thank the Reviewer for the kind and valuable comments.

Reviewer 3 Report

Neglected tropical diseases affect the lives of a billion people worldwide. Among them, the parasitic infections caused by protozoan parasites of the Trypanosomatidae family have a huge impact on human health. Leishmaniasis, caused by Leishmania spp., is an endemic parasitic disease in over 88 countries and is closely associated with poverty. Although significant advances have been made in the treatment of leishmaniasis over the last decade, currently available chemotherapy is far from satisfactory and finding efficient and safe antileishmanial agents is a need.

This manuscript is an interesting review presenting various data regarding the pysico-chemical characteristics, the mechanism of action as well as the mechanism responsible for the reduced toxicity of liposomal AmB formulations. In addition, it presents recent data on the preparation of more effective liposomal AmB formulations. Moreover, the manuscript is well written and easy to understand which is a very good point.

Author Response

Authors’ response: The authors thank the Reviewer for the revision and the kind comments.

Round 2

Reviewer 1 Report

The authors have done a good job addressing the issues now.